# Association of Advanced Lipoprotein Subpopulation Profiles with Insulin Resistance and Inflammation in Patients with Type 2 Diabetes Mellitus

**DOI:** 10.3390/jcm12020487

**Published:** 2023-01-06

**Authors:** Ahmed Bakillah, Khamis Khamees Obeid, Maram Al Subaiee, Ayman Farouk Soliman, Mohammad Al Arab, Shahinaz Faisal Bashir, Arwa Al Hussaini, Abeer Al Otaibi, Sindiyan Al Shaikh Mubarak, Jahangir Iqbal, Ali Ahmed Al Qarni

**Affiliations:** 1King Abdullah International Medical Research Center (KAIMRC), Riyadh 11481, Saudi Arabia; 2King Saud bin Abdulaziz University for Health Sciences (KSAU-HS), Riyadh 14611, Saudi Arabia; 3King Abdulaziz Hospital, Ministry of National Guard-Health Affairs (MNG-HA), Riyadh 11426, Saudi Arabia

**Keywords:** diabetes, dyslipidemia, insulin resistance, inflammation, LPIR, GlycA, lipoprotein subpopulations, lipids, hypertension, coronary artery disease

## Abstract

Plasma lipoproteins exist as several subpopulations with distinct particle number and size that are not fully reflected in the conventional lipid panel. In this study, we sought to quantify lipoprotein subpopulations in patients with type 2 diabetes mellitus (T2DM) to determine whether specific lipoprotein subpopulations are associated with insulin resistance and inflammation markers. The study included 57 patients with T2DM (age, 61.14 ± 9.99 years; HbA1c, 8.66 ± 1.60%; mean body mass index, 35.15 ± 6.65 kg/m^2^). Plasma lipoprotein particles number and size were determined by nuclear magnetic resonance spectroscopy. Associations of different lipoprotein subpopulations with lipoprotein insulin resistance (LPIR) score and glycoprotein acetylation (GlycA) were assessed using multi-regression analysis. In stepwise regression analysis, VLDL and HDL large particle number and size showed the strongest associations with LPIR (R^2^ = 0.960; *p* = 0.0001), whereas the concentrations of the small VLDL and HDL particles were associated with GlycA (R^2^ = 0.190; *p* = 0.008 and *p* = 0.049, respectively). In adjusted multi-regression analysis, small and large VLDL particles and all sizes of lipoproteins independently predicted LPIR, whereas only the number of small LDL particles predicted GlycA. Conventional markers HbA1c and Hs-CRP did not exhibit any significant association with lipoprotein subpopulations. Our data suggest that monitoring insulin resistance-induced changes in lipoprotein subpopulations in T2DM might help to identify novel biomarkers that can be useful for effective clinical intervention.

## 1. Introduction

Type 2 diabetes mellitus (T2DM) is associated with striking abnormalities of plasma lipid and lipoproteins that are associated with reduced high-density lipoprotein (HDL) cholesterol, elevated triglycerides, and a predominance of small dense low-density lipoprotein (LDL) particles [1]. These abnormalities may also occur in many diabetic patients despite achieving normal target levels of LDL cholesterol with multiple lipid-lowering drugs [2]. Insulin resistance causes significant effects on lipoprotein size and subpopulation particle concentrations of circulating plasma lipoproteins [3,4,5]. Numerous studies have consistently demonstrated that atherogenic lipoprotein particle concentrations and specifically levels of triglyceride-rich lipoprotein remnants and small dense LDL are predictive of coronary artery disease (CAD) independently of other known risk factors [6,7].

There is a large body of information linking inflammatory status and dyslipidemia [8,9,10]. Inflammatory cytokines contribute to proatherogenic changes in lipid metabolism that are mainly associated with significant reduction in HLD cholesterol and impairment of its anti-inflammatory, anti-oxidant, and reverse cholesterol functions [11,12]. Moreover, systemic inflammatory markers including high sensitivity C-reactive protein (Hs-CRP) and white blood cells are closely associated with atherogenic lipoprotein subfractions in CAD patients [13].

When assessed by the conventional lipid panel, alterations of lipids and lipoproteins in diabetic dyslipidemia do not consistently provide the full information needed for comprehensive evaluation of the effects of insulin resistance and the accompanying risks of cardiovascular disease (CVD) in diabetic patients. A recent longitudinal study of 15,464 non-diabetic participants demonstrated the importance of unconventional lipid parameters for predicting the risk of diabetes underscoring the need for better characterization of a comprehensive lipid profile evaluation in diabetic patients [14]. In the past decades, several analytical assays have been developed to separate plasma lipoproteins based on particles composition, number, and size [15,16,17,18]. With the development of new assays using nuclear magnetic resonance (NMR) spectroscopy, several studies focused on the relationship between lipoprotein subfractions and cardiovascular outcomes [19,20,21,22].

Lipoprotein insulin resistance score (LPIR) is a novel composite metabolomic biomarker that captures the multidimensional effects of insulin resistance using high throughput NMR [3]. It is a weighted score of six lipoprotein particles including VLDL, LDL, and HDL particles size; and large VLDL, small LDL, and large HDL particle concentrations that are more strongly related to insulin resistance than each of individual subclasses. LPIR score has been used to assess insulin resistance and T2DM risk and was associated with T2DM incidence independently of other established risk factors for CVD [23,24].

Glycoprotein acetylation (GlycA), a robust biomarker for systemic inflammation, has been gaining increasing interest, particularly as a new surrogate marker for diabetes and CVD risk assessment [25,26,27,28]. It was initially developed as a composite measure of changes in both the number and the complexity of N-glycan side chains attached to acute phase reactant proteins [29,30]. The NMR signal for GlycA represent a measure for methyl groups found on N-acetylglucosamine residues attached to all circulating plasma proteins including alpha-1-acid glycoprotein (AGP), alpha-1-antitrypsin (AAT), alpha-1-antichymotrypsin (AACT), haptoglobin, and transferrin [26]. GlycA has been shown to predict incidence of CVD events and T2DM, independent of traditional risk factors [25,29,31,32]. 

It is well-established that combination of hyperglycemia and chronic inflammation is associated with early subclinical carotid atherosclerosis progression and increased risk of new vascular events in diabetic as well as nondiabetic subjects [33,34]. The existing recognized biomarkers hemoglobin A1c (HbA1c) and Hs-CRP are the most clinically useful surrogates for identifying the presence of both insulin resistance and inflammation in T2DM [35]. Nevertheless, these two markers might not fully explain the pathogenesis of diabetes and its vascular complications; thus, their combination with other biomarkers can add great value for early preventive targeted strategies. Currently, there is no consensus regarding the use of lipoprotein species as biomarker signature for early diagnostic of T2DM due to insufficient data consistency from community-based studies.

Our aim in this study was to examine the effects of diabetes on plasma lipoprotein subpopulations including conventional lipids in T2DM patients. Additionally, we sought to relate specific NMR-derived lipoprotein subpopulations to the newly emerging markers LPIR and GlycA. 

## 2. Materials and Methods

### 2.1. Study Population and Protocol

A total of 57 patients with T2DM were recruited from King Abdulaziz Hospital (KAH), Ministry of National Guard-Health Affairs (MNGHA) at Al-Ahsa, Kingdom of Saudi Arabia between January 2020 and April 2021. MNGHA is an internationally respected healthcare organization government-funded multispecialty accountable health system providing outstanding patient care. It integrates clinical care, training and academics with research and cutting-edge technology with well-developed information system infrastructure that reached HIMSS INFRaM-6 maturity level. The study protocol was approved by the Institutional Review Board of MNGHA (IRB protocol# IRBC/1972/18), and written informed consent was obtained from each participant. Patients were excluded from the study if, at baseline, patients met one or more of the following criteria: patients were receiving chronic renal replacement therapy (hemodialysis, peritoneal dialysis, or transplantation), had a history of active malignancy (except those with basal cell carcinoma) within the last five years (prostatic cancer within the last two years), systemic lupus erythematosus and other autoimmune diseases that may affect kidney function, history of type 1 diabetes, acute infection or fever, pregnancy, chronic viral hepatitis or HIV infection, and current unstable cardiac disease. The following standard methods and definitions were adopted: Diabetes: subjects with a history of T2DM on medication, or HbA1c ≥ 6.5%, or fasting glucose ≥ 126 mg/dL (≥7 mmol/L). The family history of T2DM was defined as any first-degree relative being diagnosed with T2DM. Dyslipidemia: subjects with history of dyslipidemia on medication, or fasting lipid profile with total cholesterol > 200 mg/dL, or LDL > 70 mg/dL. Hypertension: subjects with systolic blood pressure ≥ 140 mmHg or diastolic blood pressure ≥ 90 mmHg and under antihypertensive medication use. CKD: subjects with eGFR < 90 mL/min, using modification of diet in renal disease (MDRD) equation or proteinuria (≥2+ on urine dipstick).

### 2.2. Quantification of Plasma Lipoprotein Subpopulations and Markers of Insulin Resistance (LPIR) and Inflammation (GlycA)

Fasting blood samples were collected in the morning, after a minimum of 12 h of fasting, into EDTA-containing tubes and centrifuged at 4 °C at 3000 rpm for 10 min to separate the plasma for biochemical tests. Samples from T2DM patients and healthy subjects were aliquoted and stored at −80 °C until further analysis. Patient medical history, demographic, and laboratory parameters were collected from the electronic medical records BESTCare database. VLDL, LDL, and HDL concentrations and size were quantified by NMR spectroscopy (LabCorp, Burlington, VA, USA). LPIR and GlycA, as well as extensive lipid profiles, were assessed by NMR lipoProfile^®^ assay (LabCorp, Burlington, VA, USA). We included a random sample of healthy controls (N = 13) to generate internal reference of normal values range for NMR-derived lipoproteins subpopulations testing including relevant lipid parameters and LPIR and GlycA markers.

### 2.3. Statistical Analysis

Statistical analyses were conducted using SPSS software version 24 (IBM Corp., Armonk, NY, USA). Normal distribution of the data was evaluated using Kolmogorov–Smirnov test. Continuous variables with normal distribution were presented as means ± SD, and non-normally distributed variables were reported as medians (IQR). Categorical variables were presented as frequencies and percentages. Independent student’s t-test was used to compare means. Associations between lipoprotein subpopulations, LPIR, GlycA, and other variables were assessed using the non-parametric Spearman’s correlation test. Stepwise analysis was performed to determine associations between PLIR, GlycA, HbA1c, and Hs-CRP as dependent variables and all lipoprotein subpopulations as independent variables. Multiple regression analysis was performed to evaluate the association between lipoprotein subpopulations and other dependent variables using adjusted-based models for covariates assessment, including factors such as age, sex, weight, height, BMI, hypertension, stroke, CAD, dyslipidemia, platelet count, creatinine, total cholesterol, LDLc, HDLc, triglycerides, and HbA1c. All adjusted β-coefficients were accompanied by approximate 95% confidence limits. Two-sided tests with *p*-values < 0.05 were considered statistically significant.

## 3. Results

### 3.1. Baseline Characteristics of the Study Subjects with T2DM

The patient characteristics of the study subjects with T2DM are detailed in Table 1. Mean patient age was 61.14 ± 9.99 years (female 44%). Ninety-five percent (95%) had hypertension, 79% had dyslipidemia and 14% had CAD. Mean HbA1c levels were 8.66± 1.60%. The majority of subjects were overweight (mean BMI = 35.15 ± 6.65 kg/m^2^). All subjects were Saudi citizens with 79% and 53% family history with diabetes and cholesterol, respectively.

### 3.2. Advanced NMR Quantification of Lipoprotein Subpopulations and Markers of Insulin Resistance and Inflammation in T2DM Patients

A comprehensive NMR analysis was performed to determine various plasma lipoprotein subpopulation profiles in T2DM patients. A random sample of 13 medical-free subjects was used to determine reference range of normal levels of lipoprotein subpopulations in healthy population. Up to twelve lipoprotein fractions were quantified (Table 2). Based on particle concentrations, large VLDL particles were higher in T2DM patients compared with reference range values. Intermediate and large LDL particles were lower in T2DM patients, whereas small LDL particles were elevated in T2DM patients in comparison with reference interval values (*p* = 0.01). Total and large HDL particles were lower in T2DM patients compared with reference range values (Table 2). The mean value of LPIR score as determined by NMR was moderately higher than normal values 56.79 ± 14.99 (LabCorp reference, Low: <30–40 and High: >60–70). The mean value of GlycA was 473.04 ± 90.87 µmol/L (≥400 µmol/L is indicative of greater inflammation related to CVD risk)

Based on the size of lipoprotein particles, average median size of both LDL and HDL particles size was slightly reduced in T2DM patients, whereas the size of VLDL particles was slightly elevated (Table 2). NMR-extended lipid panel showed a decrease in total cholesterol, LDL and HDL, whereas triglyceride levels were elevated as compared with reference range values (Table 2).

### 3.3. Correlation between Different Plasma Lipoprotein Subpopulations inT2DM Subjects

Correlations between NMR-derived lipoprotein subpopulations particles and size are listed in Table 3. Degree of correlations among various lipoproteins are modest to high correlations based on Spearman correlation analysis (rho = 0.3–0.9; Table 3)

### 3.4. Correlation of Conventional Lipid Measurements and Lipoprotein Subpopulation Profiles with PLIR Score, GlycA, HbA1c and Hs-CRP

Spearman correlation analysis revealed that among all lipoprotein subpopulation particles, LPIR was positively correlated with the number of large and medium VLDL particles and small LDL particles (Table 4; *p* = 0.0001 and *p* = 0.002, respectively), and negatively correlated with large LDL and HDL particles (Table 4; *p* = 0.001 and *p* = 0.0001, respectively). In contrast, there was no significant correlation between GlycA and lipoprotein particle concentrations. Based on the particle size of lipoproteins, LPIR positively correlated with VLDL particle size (*p* = 0.0001) and negatively correlated with LDL and HDL size (*p* = 0.0001). HbA1c levels positively correlated with LDL-c and large LDL particles (*p* = 0.02). Extended lipid panel (ELP) analysis demonstrated that LPIR positively correlated with VLDL-c, LDLc, and triglycerides (Table 4; *p* = 0.001, *p* = 0.0001, and *p* = 0.0001, respectively) and negatively correlated with HDLc (*p* = 0.0001). Among conventional measurements for lipids, LPIR score positively correlated with triglyceride levels (*p* = 0.016); whereas GlycA positively correlated with total cholesterol and LDLc levels (Table 4; *p* = 0.014 and *p* = 0.028, respectively). There was no significant correlation between Hs-CRP and various lipoprotein subpopulations and conventional lipid parameters. Additionally, fasting glucose was positively correlated with small LDL particles concentration, VLDLc, triglycerides and non-HDLc as calculated by the NMR-extended plasma lipid panel (Table 4).

### 3.5. Association of Lipoprotein Subpopulation Profiles with Glucose Control Status and Inflammation Parameters

A stepwise multiple regression was used for identification of optimal subset of predictors with estimated coefficients which best predict the value of lipoprotein subpopulation association with selected variables. LPIR, GlycA, HbA1c, and Hs-CRP were individually entered as dependent variables into stepwise regression models (Models A–D), and all lipoprotein subpopulations (number and size) were used as independent variables (Table 5). The analysis revealed that the number and size of VLDL and HDL particles exhibited stronger associations with LPIR (Model A, R^2^ = 0.96) than other individual lipoprotein subpopulations. In contrast, small VLDL and HDL particles had the greatest associations with GlycA (Model B, R^2^ = 0.19). Additionally, LDL particles number had the greatest association with HbA1c (Model C, R^2^ = 0.09) while HDL particles number had the greatest association with Hs-CRP (Model D, R^2^ =0.11).

To further analyze the independent relationship of LPIR score, GlycA, and other parameters with different lipoprotein subpopulations, multi-regression analysis was performed for each dependent variable after adjusting models for independent variables such as age, sex, weight, height, systolic blood pressure, diastolic blood pressure, platelets count, creatinine, total cholesterol, LDLc, HDLc, triglycerides, hypertension, CAD, stroke, dyslipidemia, and BMI. Among all lipoprotein subpopulations, large VLDL particles were positively associated with LPIR score (Table 6; β = 0.90, *p* = 0.0001 and R^2^_(model)_ = 0.75); whereas small VLDL particles were negatively associated with LPIR (Table 6; β= −0.78, *p* = 0.011 and R^2^_(model)_ = 0.51) after adjustment of model with potential confounding variables. In addition, small LDL particles were independently and positively associated with LPIR score (Table 6; β = 0.34, *p* = 0.035 and R^2^_(model)_ = 0.77). In contrast, VLDL size was independently and positively associated with LPIR (Table 6; β = 0.89, *p* = 0.0001 and R^2^_(model)_ = 0.85), whereas HDL and LDL particle size were independently and negatively associated with LPIR (Table 6; β = −0.45, *p* = 0.04, R^2^_(model)_ = 0.45 and β = −0.88, *p* = 0.04, Model R^2^ = 0.37, respectively).

## 4. Discussion

Lipoprotein abnormalities have been associated with increased risk of CVD in diabetic patients [27,36,37]. Gaining better understanding of exact mechanisms underlying lipoprotein disturbance during the progression of diabetes is crucial to tackle early diabetic vascular complications beyond traditional glycemic control [38,39]. Numerous procedures have been extensively used to study in depth lipoprotein profile changes including NMR [17,40,41]. However, results from different ethnic populations, heterogeneous pool cohorts, patient’s age, and the nature and duration of disease were not consistent [27,42,43,44]. During the past few decades, several lipid lowering drugs have been evaluated in many studies with focus not only on the reduction in LDLc, but also on LDL particles size extending its clinical value beyond the information collected from a traditional lipid panel [45,46,47,48,49]. The increase in small dense LDL particles was established as the main feature of the pathophysiology of diabetic dyslipidemia leading to atherosclerotic CVD [50,51,52]. However, despite optimal current lipid lowering therapies including statins, many diabetic patients remain at high residual risk of CVD due to the effect of other factors including increased hepatic secretion of large triglyceride-rich VLDL, impaired clearance of VLDL, and low HDL particles [53,54].

While there are some challenges and limitations for universal advanced lipoprotein profiling to be resolved, data generated in this study and from other groups consistently indicate that comprehensive NMR-derived lipoproteins analysis is a reliable and powerful tool that can expand diagnostic value and disease management when interpreting results of lipid panel and lipoproteins disturbance in T2DM patients. Most of published studies on advanced assessment of lipoproteins species by NMR technique have been focusing on the prediction of T2DM in apparently healthy populations [4,22,55,56,57,58,59]. In this study we provide a comprehensive examination of the effects of insulin resistance on NMR-determined lipoprotein subpopulation profiles in T2DM patients. NMR generates unique spectra by simultaneously quantifying different lipoprotein subpopulations (sizes and concentrations) in single plasma run along with two emerging potential markers LPIR score and GlycA. Consistent with previous reports, our study showed that T2DM patients had a notable increase in number of large VLDL and small LDL particles as compared with reference values of healthy subjects (Table 2) and other published data in healthy populations [22,31,58,60]. Additionally, T2DM patients had a notable decrease in the number of large HDL and LDL particles. While VLDL particles size remained unchanged, the size of HDL and LDL particles was reduced in T2DM patients (Table 2) compared with reference values of healthy controls. These changes were associated with marked elevation of the insulin resistance score LPIR and moderate increase in the systemic inflammatory marker GlycA (Table 2).

Our results reproduce and support the observations by Garvey and colleagues demonstrating that progressive changes in specific lipoprotein classes occur during diabetes as insulin resistance becomes more severe [60]. These findings were further confirmed by correlation analysis which documented that number and size of VLDL particles were positively correlated with LPIR, whereas number and size of HDL and LDL were negatively correlated with LPIR (Table 4). Interestingly, these findings were not apparent with conventional parameters of glucose metabolism such as HbA1c and fasting glucose (Table 4). Furthermore, our data also revealed significant correlations of LPIR with NMR-extended lipid panel (ELP) including triglycerides but not with conventional lipids measurement (Table 4).

Despite the relatively small size of our study cohort, our results support findings reported from large trials such as the Multi-Ethnic Study of Atherosclerosis (MESA) [24], the Women’s Health Study (WHS) [23], and the Prevention of Renal End Stage Disease (PREVEND) [61] demonstrating a robust association of LPIR score with incident T2DM independent of established risk factors.

Inflammation and dyslipidemia are well-established cardiovascular risk factors and strongly associated with each other. The presence of dyslipidemia by itself can trigger the inflammatory process through alterations of lipoproteins such as oxidized LDL [62]. Interestingly, we found that there was no correlation between lipoprotein subpopulations and inflammatory markers GlycA and Hs-CRP (Table 4). In addition, conventional lipids such as total cholesterol and LDLc were positively correlated with GlycA but not with Hs-CRP (Table 4). This finding is not in agreement with a study reporting a close association between atherogenic lipoprotein subfractions and systemic inflammatory markers [13]. Additionally, a recent study showed an independent association between NMR-derived remnant lipoproteins and Hs-CRP with risk of atherosclerotic CVD [63]. This discrepancy may be explained by the difference between the study populations in which the authors used CVD patients compared with T2DM patients in this study. It is also more conceivable that inflammation plays a more important role during plaque instability of advanced atherosclerotic lesions [64,65]. Another plausible explanation of lack of correlation between lipoprotein subpopulations and hs-CRP can be related to high intra-individual variability of Hs-CRP values in our cohort based on one single measurement at baseline. Interestingly, we found that GlycA positively correlated with Hs-CRP and BMI (Spearman rho; r = 0.345, *p* = 0.019 and r = 0.277, *p* = 0.037, respectively). Although, there is a modest association between GlycA and Hs-CRP, it is worth noting that these two markers reflect distinct inflammatory metabolic processes. GlycA is a composite marker that involves multiple circulating glycosylated proteins, whereas Hs-CRP reflects a single acute-phase protein [32,66]. A study of pooled cohort from the Dallas Heart study (DHS) and Multi-Ethnic Study of Atherosclerosis (MESA) demonstrated differential association of GlycA and Hs-CRP with CVD; GlycA strongly predicted incident of myocardial infraction, and Hs-CR strongly predicted ischemic stroke [67]. However, the PREVEND study demonstrated that GlycA, but not Hs-CRP, was significantly associated with incident T2DM [68].

Stepwise regression analysis demonstrated that the size and number of large VLDL and HDL particles were the strongest predictors for insulin resistance score LPIR followed by small VLDL particles (Table 5, Model A). In contrast, the other dependent variables such as GlycA, HbA1c, and Hs-CRP exhibited weak associations with lipoprotein subpopulations that were explained with (10–20%) variance only (Table 5, models C-D). Furthermore, multi-regression analysis showed that LPIR score association with large VLDL particles number remained significant after adjusting the model with other potential confounding parameters (Table 6). This finding is consistent with the role of large VLDL particles in the initiation of early sequences of events in T2DM that generate atherogenic lipoprotein remnants, small dense LDL, and small HDL particles [69,70]. Additionally, all sizes of lipoprotein particles were independently associated with LPIR in the adjusted multi-regression models (Table 6). VLDL size positively associated with LPIR score, whereas size of HDL and LDL particles exhibited negative association with LPIR score. It is unlikely that use of certain drugs in our cohort could have altered LPIR score and subsequently affected the interpretation of the observed associations with lipoprotein subpopulations. In fact, studies have reported that use of lipid-modifying drugs did not have any substantial effect on LPIR score [3,4].

Our findings are also consistent with results from the PREVEND study reporting close association of triglyceride-rich lipoprotein particles and LDL particles with incident T2DM [57]. Furthermore, the JUPITER study demonstrated evidence that the LPIR score was indeed positively associated with incident diabetes in initially healthy T2DM subjects, even though treatment by rosuvastatin resulted in significant reduction in LDLc and triglycerides levels without any apparent effect on LPIR score; suggesting that VLDL and LDL particles were most likely shifted toward the larger size of VLDL and smaller size of LDL particles [4]. Another recent large study provided a comprehensive analysis of lipoproteins subpopulations showing higher T2DM risk among participants with higher concentrations of large VLDL particles, lower concentrations of large HDL particles, and smaller mean HDL particle size [56] which is again consistent with our findings.

Several limitations of the current observational study must be considered. First, this is a retrospective study of a relatively small sample size of overweight/obese T2DM subjects recruited from a single center which may have limited the power to detect weak correlations among measured variables; however, the sample size of T2DM participants in this study was sufficient to demonstrate strong correlations between specific lipoprotein subpopulations and the index of insulin resistance LPIR. Second, no conclusion regarding causality can be drawn regarding reported associations in this observational study for which duration of T2DM was not recorded. Third, the majority of participants were on various medications which may have biased the value of associations between relevant variables. The current study did not assess the effects of lipid-lowering drugs such as statins which could affect lipoproteins distribution; however, it is unlikely that these drugs would directly impact LPIR association with atherogenic lipoprotein subpopulations. Lastly, the study population represents a community-based cohort of relatively obese T2DM participants (all Saudi citizens) from the Eastern region of the Kingdom of Saudi Arabia; therefore, these results cannot be generalized to other ethnic groups. However, through a comparison with results from other ethnic groups this study may provide better understanding for the relationship between lipoprotein subpopulations and insulin resistance in T2DM patients.

## 5. Conclusions

In summary, advanced NMR-derived lipoproteins showed that VLDL and HDL were the strongest predictors for insulin resistance score in T2DM patients. Large VLDL particles number and size were positively associated with LPIR, whereas size of LDL and HDL particles were inversely associated with LPIR, but not with HbA1c levels. In contrast, the systemic inflammation marker GlycA was positively associated with small LDL particles only. Interestingly, the number of particles and size of the atherogenic lipoprotein subpopulations seem to be more robust in predicting insulin resistance than systemic inflammation in T2DM patients. Large prospective longitudinal studies are warranted for further investigations to demonstrate clinical superiority of advanced lipoprotein profiling. This will provide the basis for the identification of potential biomarkers-based lipoprotein subpopulations that can be measured to predict and prevent T2DM.

## Figures and Tables

**Table 1 jcm-12-00487-t001:** Characteristics of the study subjects.

Characteristics	Study Cohort (N = 57)
Age (years) *	
Mean (SD)	61.14 (9.99)
Height (cm) *	
Mean (SD)	160.58 (8.60)
BMI (kg/m^2^) *	
Mean (SD)	35.15 (6.65)
Systolic BP (mmHg)	
Median (IQR)	145.00 (123.50–153.00)
Diastolic BP (mmHg)	
Median (IQR)	71.00 (61.50–81.00)
HbA1c (%) *	
Mean (SD)	8.66 (1.60)
Total Cholesterol (mg/dL)	
Mean (SD)	162.03(51.43)
LDLc (mg/dL)	
Mean (SD)	99.38 (38.74)
HDLc (mg/dL)	
Mean (SD)	45.24 (25.13)
Triglycerides (mg/dL)	
Mean (SD)	162.97 (84.32)
Platelet Count (×10^3^/mL)	
Median (IQR)	260.00 (217.50–290.00)
Creatinine (mg/dL)	
Median (IQR)	1.46 (1.09–1.63)
Total proteins (g/L)	
Median (IQR)	70.00 (67.25–72.75)
Fasting Glucose (mg/dL)	144.14 (120.18–257.12)
Hs-CRP (mg/L)	7.60 (3.40–25.97)
Dyslipidemia, n (%)	45 (79)
Hypertension, n (%)	54 (95)
CAD, n (%)	8 (14)
Family History of Diabetes (%)	78.90
Family History of Hypertension (%)	24.60
Family History of CAD (%)	36.80
Family History of Cholesterol (%)	52.60
Family History of Stroke (%)	14.00
Insulin (%)	47.37
HMG-CoA reductase inhibitors (%)	19.30
Metformin (%)	14.03
DPP4 inhibitors (%)	14.03
Sulfonylurea (%)	10.53
Calcium channel blockers (%)	8.77
ACE inhibitors (%)	7.02
NSAID (%)	7.02
Diuretics (%)	3.51
PPI (%)	3.51

Data are presented for continuous variables as mean (standard deviation, SD) or median (interquartile range, IQR), and as frequencies (percentages) for categorical variables. *: data normally distributed; ND: not determined; BMI: body mass index; BP: blood pressure; HbA1c: hemoglobin A1c; LDLc: low density lipoprotein cholesterol; HDLc: high density lipoprotein cholesterol; Hs-CRP: high sensitivity C-reactive protein; CAD: coronary artery disease; DPP4: dipeptidyl-peptidase 4; ACE: angiotensin-converting-enzyme; NSAID: non-steroidal anti-inflammatory drugs; PPI: proton-pump inhibitor.

**Table 2 jcm-12-00487-t002:** Plasma levels of NMR-derived lipoprotein subpopulations and markers for insulin resistance and inflammation in T2DM patients.

Variables	T2DM Patients (N = 57)	NMR Internal Quality Controls (Reference Range) *
**Plasma Lipoprotein Concentration**		
VLDLCP3 (nmol/L)	35.80 [25.95–51.05]	46.20 [28.65–56.45]
VLCP3 (nmol/L)	4.40 [2.65–6.05]	2.50 [1.20–5.80]
VMCP3 (nmol/L)	8.80 [2.65–6.05]	11.90 [7.45–27.65]
VSCP3 (nmol/L)	20.80 [12.15–29.75]	20.00 [13.60–28.60]
LDLCP3 (nmol/L)	906.00 [725.00–1146.50]	1123.00 [873.50–1177.00]
IDLCP3 (nmol/L)	87.0 [39.50–157.50]	234.00 [131.00–283.00]
LLCP3 (nmol/L)	129.00 [15.50–316.00]	301.00 [155.00–490.00]
LSCP3 (nmol/L)	657.00 [533.00–816.00]	440.00 [321.50–654.00]
HDLCP3 (nmol/L)	29.50 [27.10–31.85]	35.10 [30.00–37.55]
HLCP3 (nmol/L)	5.70 [4.40–7.05]	8.80 [7.30–10.80]
HMCP3 (nmol/L)	5.40 [2.65–10.10]	8.10 [4.65–12.65]
HSCP3 (nmol/L)	18.50 [12.75–20.95]	16.90 [9.65–21.65]
**Plasma Lipoprotein Size**		
VZ3 (nm)	55.70 [50.80–59.10]	48.00 [45.50–56.75]
LZ3 (nm)	20.20 [19.70–20.75]	21.30 [20.50–21.70]
HZ3 (nm)	9.30 [9.10–9.50]	9.60 [9.50–10.00]
**Plasma-Extended Lipid Panel**		
ELP-Total cholesterol (mg/dL)	144.00 [127.00–176.50]	195.00 [165.00–213.00]
ELP-VLDLc (mg/dL)	22.00 [19.00–31.50]	17.00 [14.50–29.50]
ELP-HDLc (mg/dL)	42.00 [33.50–46.00]	54.00 [50.50–61.50]
ELP-LDLc (mg/dL)	77.00 [63.50–102.50]	116.00 [97.00–127.00]
ELP-TG (mg/dL)	120.00 [102.50–172.00]	90.00 [77.50–167.00]
ELP-nonHDLc (mg/dL)	106.00 [89.50–134.50]	138.00 [113.50–165.50]
**NMR-Derived Markers for Insulin Resistance and Inflammation**		
LPIR Score ^#^	57.00 [45.50–67.50]	33.00 [16.00–45.50]
GlycA (µmol/L)	463.00 [409.50–525.50]	393.00 [341.00–470.50]

Values are presented as median (IQR, 25th–75th percentiles); VLCP3: large very low-density lipoprotein particle concentration; VMCP3: medium very low-density lipoprotein particle concentration; VSCP3: small very low-density lipoprotein particle concentration; LDLCP3: low-density lipoprotein particle concentration; IDLCP3: intermediate low-density lipoprotein particle concentration; LLCP3: large low-density lipoprotein particle concentration; LSCP3: small low-density lipoprotein particle concentration; HDLCP3: high-density lipoprotein particle concentration; HLCP3: large high-density lipoprotein particle concentration; HMCP3: medium high-density lipoprotein particle concentration; HSCP3: small high-density lipoprotein particle concentration. VZ3: very low-density lipoprotein size; HZ3: high-density lipoprotein size; LZ3: low-density lipoprotein size; Extended lipid panel (ELP) was determined by partial least squares regression (PLS) analysis of NMR outcome; LPIR: lipoprotein insulin resistance. *: data presented are generated from random healthy subjects (n = 13) and used as reference range for comparison with T2DM group; ^#^: no-gender specific determination.

**Table 3 jcm-12-00487-t003:** Spearman rank correlation matrix among plasma NMR-derived lipoproteins concentration and size in T2DM patients.

Variables		VLDL	VLCP3	VMP3	VSP3	LDL	IDLP3	LLP3	LSP3	HLD	HLP3	HMP3	HSP3	VZ3	LZ3	HZ3
		CP3				CP3				CP3						
VLDLCP3	R	.	**0.577 ****	0.558 **	**0.804 ****	0.063	−0.1	**−0.292 ***	**0.282 ***	−0.062	**−0.410 ****	0.017	0.139	−0.107	**−0.437 ****	**−0.371 ****
	P	.	**0.0001**	0.0001	**0.0001**	0.641	0.46	**0.028**	**0.033**	0.645	**0.002**	0.897	0.301	0.428	**0.001**	**0.004**
VLCP3	R	**0.577 ****	.	**0.644 ****	0.158	0.085	−0.213	**−0.387 ****	**0.432 ****	0.074	**−0.431 ****	0.092	0.117	**0.657 ****	**−0.484 ****	−0.440 **
	P	**0.0001**	.	**0.0001**	0.24	0.528	0.111	**0.003**	**0.001**	0.584	**0.001**	0.497	0.387	**0.0001**	**0.0001**	0.001
VMP3	R	**0.558 ****	**0.644 ****	.	0.054	0.078	−0.075	-0.23	0.241	0.03	**−0.351 ****	0.144	0.075	**0.453 ****	**−0.348 ****	−0.371 **
	P	**0.0001**	**0.0001**	.	0.688	0.565	0.577	0.086	0.071	0.589	**0.007**	0.285	0.579	**0.0001**	**0.008**	0.005
VSP3	R	**0.804 ****	0.158	0.054	.	0.0001	−0.01	−0.121	0.086	−0.047	**−0.262 ***	−0.075	0.173	**−0.526 ****	−0.229	−0.205
	P	**0.0001**	0.24	0.688	.	0.999	0.939	0.371	0.527	0.727	**0.049**	0.577	0.198	**0.0001**	0.086	0.125
LDL-	R	0.063	0.085	0.078	0.0001	.	**0.405 ****	**0.575 ****	**0.650 ****	0.251	0.193	0.042	0.014	−0.044	**0.384 ****	−0.013
CP3	P	0.641	0.528	0.565	0.999	.	**0.002**	**0.0001**	**0.0001**	0.059	0.15	0.755	0.92	0.743	**0.003**	0.922
IDLP3	R	−0.1	−0.213	−0.075	−0.01	**0.405 ****	**.**	**0.278 ***	−0.113	0.038	0.146	0.26	−0.219	−0.118	**0.364 ****	**0.264 ***
	P	0.46	0.111	0.577	0.939	**0.002**	**.**	**0.037**	0.403	0.779	0.28	0.051	0.101	0.383	**0.005**	**0.047**
LLP3	R	**−0.292 ***	**−0.387 ****	−0.23	−0.121	**0.575 ****	**0.278 ***	.	−0.06	0.148	**0.552 ****	0.037	−0.087	**−0.311 ***	**0.889 ****	**0.354 ****
	P	**0.028**	**0.003**	0.086	0.371	**0.0001**	**0.037**	.	0.66	0.271	**0.0001**	0.783	0.521	**0.018**	**0.0001**	**0.007**
LSP3	R	**0.282 ***	**0.432 ****	0.241	0.086	**0.650 ****	−0.113	−0.06	.	0.099	**−0.263 ***	−0.124	0.143	0.184	**−0.322 ***	** *−* ** **0.435 ****
	P	**0.033**	**0.001**	0.071	0.527	**0.0001**	0.403	0.66	.	0.464	**0.048**	0.358	0.29	0.171	**0.015**	**0.001**
HLD-	R	−0.062	0.074	0.073	−0.047	0.251	0.038	0.148	0.099	.	0.187	0.128	**0.517 ****	0.165	0.174	−0.126
CP3	P	0.645	0.584	0.589	0.727	0.059	0.779	0.271	0.464	.	0.163	0.341	**0.0001**	0.22	0.196	0.351
HLP3	R	**−0.410 ****	**−0.431 ****	**−0.351 ****	−0.262 *	0.193	0.146	**0.552 ****	**−0.263 ***	0.187	.	0.169	-0.223	−0.175	**0.685 ****	**0.835 ****
	P	**0.002**	**0.001**	**0.007**	0.049	0.15	0.28	**0.0001**	**0.048**	0.163	.	0.21	0.095	0.194	**0.0001**	**0.0001**
HMP3	R	0.017	0.092	0.144	−0.075	0.042	0.26	0.037	−0.124	0.128	0.169	.	**−0.683 ****	0.226	0.141	**0.281 ***
	P	0.897	0.497	0.285	0.577	0.755	0.051	0.783	0.358	0.341	0.21	.	**0.0001**	0.092	0.296	**0.035**
HSP3	R	0.139	0.117	0.075	0.173	0.014	−0.219	−0.087	0.143	**0.517 ****	**−0.223**	**−0.683 ****	.	−0.04	−0.186	**−0.511 ****
	P	0.301	0.387	0.579	0.198	0.92	0.101	0.521	0.29	**0.0001**	**0.095**	**0.0001**	.	0.767	0.166	**0.0001**
VZ3	R	−0.107	**0.657 ****	**0.453 ****	**−0.526 ****	−0.044	−0.118	**−0.311 ***	0.184	0.165	−0.175	0.226	−0.04	.	**−0.270 ***	−0.192
	P	0.428	**0.0001**	**0.0001**	**0.0001**	0.743	0.383	**0.018**	0.171	0.22	0.194	0.092	0.767	.	**0.042**	0.153
LZ3	R	**−0.437 ****	**−0.484 ****	**−0.348 ****	−0.229	**0.384 ****	**0.364 ****	**0.889 ****	**−0.322 ***	0.174	**0.685 ****	0.141	−0.186	**−0.270 ***	.	**0.592 ****
	P	**0.001**	**0.0001**	**0.008**	0.086	**0.003**	**0.005**	**0.0001**	**0.015**	0.196	**0.0001**	0.296	0.166	**0.042**	.	**0.0001**
HZ3	R	**−0.371 ****	**−0.440 ****	**−0.371 ****	−0.205	−0.013	**0.264 ***	**0.354 ****	**−0.435 ****	−0.126	**0.835 ****	**0.281 ***	**−0.511 ****	−0.192	**0.592 ****	.
	P	**0.004**	**0.001**	**0.005**	0.125	0.922	**0.047**	**0.007**	**0.001**	0.351	**0.0001**	**0.035**	**0.0001**	0.153	**0.0001**	.

Results are expressed as Spearman’s rho coefficient R and *p*-values for 2-tailed significance: *, *p* < 0.05; and **, *p* < 0.001 VLDLCP3: very-low density lipoprotein and chylomicron particle concentration; VLCP3: very-low density lipoprotein particle concentration; VMCP3: medium very low-density lipoprotein particle concentration; VSCP3: small very low-density lipoprotein particle concentration; LDLCP3: low-density lipoprotein particle concentration; IDLCP3: intermediate low-density lipoprotein particle concentration; LLCP3: large low-density lipoprotein particle concentration; LSCP3: small low-density lipoprotein particle concentration; HDLCP3: high-density lipoprotein particle concentration; HLCP3: large high-density lipoprotein particle concentration; HMCP3: medium high-density lipoprotein particle concentration; HSCP3: small high-density lipoprotein particle concentration; VZ3: very low-density lipoprotein size; HZ3: high-density lipoprotein size; LZ3: low-density lipoprotein size.

**Table 4 jcm-12-00487-t004:** Spearman correlation analysis between plasma lipids and lipoproteins subpopulations with insulin resistance and inflammation markers in T2DM patients.

Variables	LPIR	GlycA	HbA1c	Hs-CRP	F-Gluc
**Plasma Lipoprotein Concentration**					
VLDLCP3	0.221 (0.101)	0.079 (0.562)	0.214 (0.119)	0.074 (0.630)	0.263 (0.068)
VLCP3	**0.797 *** ** **(0.0001)**	−0.116 (0.394)	0.190 (0.168)	0.037 (0.809)	0.219 (0.131)
VMCP3	**0.500 *** ** **(0.0001)**	0.019 (0.888)	−0.002 (0.987)	0.046 (0.764)	0.158 0.277
VSCP3	−0.146 (0.283)	0.115 (0.400)	0.244 (0.076)	0.036 (0.814)	0.197 (0.175)
LDLCP3	−0.007 (0.958)	0.104 (0.447)	**0.320 * ** **(0.018)**	−0.045 (0.767)	0.248 (0.086)
IDLCP3	−0.232 (0.086)	0.099 (0.469)	0.121 (0.382)	0.080 (0.603)	−0.005 (0.970)
LLCP3	**−0.449 ** ** **(0.001)**	0.103 (0.451)	**0.305 * ** **(0.025)**	0.136 (0.375)	0.022 (0.881)
LSCP3	**0.400 ** ** **(0.002)**	0.136 (0.319)	0.167 (0.226)	−0.057 (0.708)	**0.285 * ** **(0.047)**
HDLCP3	0.093 (0.495)	-0.055 (0.690)	−0.105 (0.451)	−0.236 (0.118)	−0.047 (0.748)
HLCP3	**−0.627 *** ** **(0.0001)**	0.135 (0.321)	0.032 (0.819)	0.050 (0.742)	−0.009 (0.952)
HMCP3	0.034 (0.805)	0.135 (0.322)	0.147 (0.288)	0.144 (0.344)	−0.139 0.341
HSCP3	0.176 (0.196)	-0.203 (0.133)	−0.251 (0.068)	−0.285 (0.058)	0.001 (0.993)
**Plasma Lipoprotein Size**					
VZ3	**0.781 *** ** **(0.0001)**	−0.144 (0.288)	−0.053 (0.704)	−0.005 (0.972)	−0.011 (0.942)
LZ3	**−0.550 *** ** **(0.0001)**	0.093 (0.498)	0.225 (0.101)	0.112 (0.462)	−0.060 (0.680)
HZ3	**−0.642 *** ** **(0.0001)**	0.145 (0.286)	0.116 (0.405)	0.069 (0.652)	0.005 (0.972)
**Extended Plasma Lipid Panel**					
ELP-Total cholesterol	−0.168 (0.216)	0.211 (0.119)	**0.315 * ** **(0.020)**	0.060 (0.698)	0.239 (0.098)
ELP-VLDLc	**0.611 *** ** **(0.0001)**	0.046 (0.734)	0.193 (0.162)	0.112 (0.466)	**0.295 * ** **(0.040)**
ELP-HDLc	**−0.610 *** ** **(0.0001)**	0.136 (0.319)	0.063 (0.651)	−0.022 (0.886)	0.005 (0.975)
ELP-LDLc	**0.268 * ** **(0.045)**	0.261 (0.052)	**0.375 **** (**0.005**)	0.132 (0.387)	0.255 (0.076)
ELP-TG	**0.600 *** ** **(0.0001)**	0.053 (0.696)	0.173 (0.210)	0.119 (0.438)	**0.298 * ** **(0.037)**
ELP-nonHDLc	−0.036 (0.792)	0.201 (0.137)	**0.354 ** ** **(0.009)**	0.050 (0.746)	**0.287 * ** **(0.045)**
**Conventional Plasma Lipid Panel**					
Total cholesterol	−0.189 (0.213)	**0.364 * ** **(0.014)**	0.026 (0.864)	0.093 (0.585)	−0.003 (0.984)
LDLc	−0.259 (0.086)	**0.327 * ** **(0.028)**	0.091 (0.551)	0.082 (0.628)	−0.038 (0.813)
HDLc	−0.026 (0.865)	−0.092 (0.551)	0.261 (0.087)	0.181 (0.292)	0.111 90.494)
Triglycerides	**0.363 * ** **(0.016)**	0.226 (0.141)	0.071 (0.648)	0.159 (0.354)	0.113 (0.489)
ApoB	0.008 (0.951)	0.224 (0.098)	**0.373 ** ** **(0.005)**	0.027 (0.859)	0.279 (0.052)

Results are expressed as Spearman’s rho coefficient R and (*p*-value) for 2-tailed significance: *, *p* < 0.05; **, *p* < 0.001 and ***, *p* < 0.0001. LPIR: lipoprotein insulin resistance; GlycA: glycoprotein acetylation; HbA1c: hemoglobin A1c; Hs-CRP: high sensitivity C-reactive protein; F-Gluc: fasting glucose; VLDL: very low-density lipoprotein; VLCP3: large very low-density lipoprotein particle concentration; VMCP3: medium very low-density lipoprotein particle concentration; VSCP3: small very low-density lipoprotein particle concentration; LDLCP3: low-density lipoprotein particle concentration; IDLCP3: intermediate low-density lipoprotein particle concentration; LLCP3: large low-density lipoprotein particle concentration; LSCP3: small low-density lipoprotein particle concentration; HDLCP3: high-density lipoprotein particle concentration; HLCP3: large high-density lipoprotein particle concentration; HMCP3: medium high-density lipoprotein particle concentration; HSCP3: small high-density lipoprotein particle concentration; VZ3: very low-density lipoprotein size; HZ3: high-density lipoprotein size; LZ3: low-density lipoprotein size; ApoB: apolipoprotein B; extended lipid panel (ELP) was determined by partial least squares regression (PLS) analysis of NMR outcome.

**Table 5 jcm-12-00487-t005:** Stepwise regression analysis of the association of lipoprotein subpopulations with LPIR, GlycA, HbA1c, and Hs-CRP in T2DM patients.

**Model. A ** **(R^2^ = 0.960) ** **LPIR ***	Unstandardized Coefficients		Standardized Coefficients	t	*p*-Value
	β Coefficient	Standard Error	β Coefficient		
(Constant)	73.947	15.818		4.675	0.0001
VZ3	1.416	0.136	0.684	10.434	**0.0001**
HZ3	−9.849	1.509	−0.279	−6.528	**0.0001**
VLCP3	0.806	0.297	0.147	2.714	**0.0001**
HLCP3	−1.654	0.333	−0.210	−4.971	**0.0001**
VSCP3	0.118	0.048	0.119	2.446	**0.0180**
**Model. B. ** **(R^2^ = 0.190) ** **GlycA ***	Unstandardized Coefficients		Standardized Coefficients	t	*p*-Value
	β Coefficient	Standard Error	β Coefficient		
(Constant)	480.695	34.819		13.806	0.0001
VSCP3	2.062	0.742	0.344	2.780	**0.008**
HSCP3	−3.316	1.646	−0.249	−2.015	**0.049**
**Model. C ** **(R^2^ = 0.092) ** **HbA1c ***	Unstandardized Coefficients		Standardized Coefficients	t	*p*-Value
	β Coefficient	Standard Error	β Coefficient		
(Constant)	7.340	0.607		12.089	0.0001
LDLCP3	0.001	0.001	0.304	2.299	**0.026**
**Model. D ** **(R^2^ = 0.112) ** **Hs-CRP ***	Unstandardized Coefficients		Standardized Coefficients	t	*p*-Value
	β Coefficient	Standard Error	β Coefficient		
(Constant)	−12.256	15.689		−0.781	0.439
HLCP3	6.010	2.586	0.334	2.324	**0.025**

*: dependent variables: LPRI (Model A), GlycA (Model B), HbA1c (Model C), and Hs-CRP (Model D). All the twelve lipoprotein subpopulations (VLDL, HDL, and LDL) particle concentration and size were included as independent variables in each regression model (A–D) to determine the strongest predictor for the association between variables. R Square is the proportion of variance between variables in the regression model. Data are expressed as standardized regression coefficient β at 95% confidence intervals. VLDL: very low-density lipoprotein; VLCP3: large very low-density lipoprotein particle concentration; VSCP3: small very low-density lipoprotein particle concentration; LDLCP3: low-density lipoprotein particle concentration; HDLCP3: high-density lipoprotein particle concentration; HLCP3: large high-density lipoprotein particle concentration; small high-density lipoprotein particle concentration; VZ3: very low-density lipoprotein size; HZ3: high-density lipoprotein size.

**Table 6 jcm-12-00487-t006:** Multiple linear regression analysis of the association of lipoprotein subpopulations with glycemic and inflammation parameters in T2DM patients.

Independent Variables	LPIR *		GlycA *		HbA1c *		Hs-CRP *	
	β	*p*-Value	β	*p*-Value	β	*p*-Value	β	*p*-Value
**Advanced Lipoprotein Particle Concentration**								
VLCP3	**0.900 ** **(R^2^ = 0.755)**	**0.0001**	−0.046 (R^2^ = 0.719)	0.781	0.148 (R^2^ = 0.526)	0.489	0.142 (R^2^ = 0.615)	0.691
VMCP3	0.340 (R^2^ = 0.372)	0.215	0.303 (R^2^ = 0.759)	0.081	0.145 (R^2^ = 0.523)	0.538	0.247 (R^2^ = 0.634)	0.355
VSCP3	**−0.778 ** **(R^2^ = 0.512)**	**0.011**	0.302 (R^2^ = 0.746)	0.148	0.138 (R^2^ = 0.520)	0.623	−0.051 (R^2^ = 0.611)	0.892
HLCP3	−0.444 (R^2^ = 0.432)	0.061	0.048 (R^2^ = 0.719)	0.765	−0.004 (R^2^ = 0.514)	0.984	0.225 (R^2^ = 0.618)	0.602
HMCP3	−0.035 (R^2^ = 0.321)	0.877	0.130 (R^2^ = 0.729)	0.365	−0.018 (R^2^ = 0.514)	0.925	0.324 (R^2^ = 0.677)	0.111
HSCP3	0.046 (R^2^ = 0.322)	0.872	−0.049 (R^2^ = 0.719)	0.788	−0.190 (R^2^ = 0.529)	0.426	0.208 (R^2^ = 0.633)	0.367
IDLCP3	−0.308 (R^2^ = 0.359)	0.287	−0.213 (R^2^ = 0.736)	0.251	0.171 (R^2^ = 0.526)	0.489	0.269 (R^2^ = 0.636)	0.340
LLCP3	−0.261 (R^2^ = 0.346)	0.389	0.228 (R^2^ = 0.737)	0.238	0.176 (R^2^ = 0.525)	0.494	0.116 (R^2^ = 0.614)	0.708
LSCP3	0.442 (R^2^ = 0.420)	0.079	**0.336 ** **(R^2^ = 0.775)**	**0.035**	0.039 (R^2^ = 0.515)	0.860	0.084 (R^2^ = 0.613)	0.763
**Advanced Lipoprotein Particle Size**								
VZ3	**0.890 ** **(R^2^ = 0.847)**	**0.0001**	−0.149 (R^2^ = 0.733)	0.305	−0.036 (R^2^ = 0.515)	0.644	0.149 (R^2^ = 0.620)	0.554
HZ3	**−0.455 ** **(R^2^ = 0.453)**	**0.040**	−0.060 (R^2^ = 0.720)	0.688	0.135 (R^2^ = 0.526)	0.492	−0.185 (R^2^ = 0.620)	0.556
LZ3	**−0.484 ** **(R^2^ = 0.457)**	**0.037**	−0.008 (R^2^ = 0.718)	0.961	0.133 (R^2^ = 0.524)	0.519	0.123 (R^2^ = 0.616)	0.643

*: dependent variables: LPIR, GlycA, HbA1c, and Hs-CRP; R^2^: proportion of variance between variables in linear regression model. Multiple linear regression was performed individually for each lipoprotein subpopulation after adjusting the model with other independent variables such as age, sex, weight, height, systolic blood pressure, diastolic blood pressure, platelets count, creatinine, total cholesterol, LDLc, HDLc, triglycerides, hypertension, CAD, stroke, dyslipidemia, and BMI. Data are expressed as standardized regression coefficient β at 95% confidence intervals. VLDL: very low-density lipoprotein; VLCP3: large very low-density lipoprotein particle concentration; VMCP3: medium very low-density lipoprotein particle concentration; VSCP3: small very low-density lipoprotein particle concentration; LDLCP3: low-density lipoprotein particle concentration; IDLCP3: intermediate low-density lipoprotein particle concentration; LLCP3: large low-density lipoprotein particle concentration; LSCP3: small low-density lipoprotein particle concentration; HDLCP3: high-density lipoprotein particle concentration; HLCP3: large high-density lipoprotein particle concentration; HMCP3: medium high-density lipoprotein particle concentration; HSCP3: small high-density lipoprotein particle concentration; VZ3: very low-density lipoprotein size; HZ3: high-density lipoprotein size; LZ3: low-density lipoprotein size.

## Data Availability

Not applicable.

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
