# Peer review of "Association of Advanced Lipoprotein Subpopulation Profiles with Insulin Resistance and Inflammation in Patients with Type 2 Diabetes Mellitus"

_jcm, 2023, doi:10.3390/jcm12020487_

Round 1

Reviewer 1 Report

This is an interesting study regarding the potential use of a new inflammatory biomarke4 for diabetes progression. However there are several limitations including the weak design of the study. Since the number of controls is very minimal, no solid conclusions can be drawn from such findings and authors are advised to increase the number of controls.

After adding more control cases, data analysis can be considered.

Also the design of the study is not well explained in the methods section.

For results table 1 is not informative if data for control group is not represented. 

Similarly for table 4 the comparison for sex differences is even more faulty because of the small sample size and lack of comparison. Stratified the data in this case doesn't inform about the strength of association.

For discussion, authors are advised to elaborate more on the results of the study in comparison to other studies even if results are not in concordance. Almost all studies referenced agree with obtained results.

Author Response

Responses to Reviewer’s #1 Comments

We thank the reviewer for his/her excellent constructive review and very useful comments/suggestions. This is an interesting study regarding the potential use of a new inflammatory biomarke4 for diabetes progression. However, there are several limitations including the weak design of the study. Since the number of controls is very minimal, no solid conclusions can be drawn from such findings and authors are advised to increase the number of controls. After adding more control cases, data analysis can be considered. Also, the design of the study is not well explained in the methods section.

We are glad to hear that you found our study interesting.

We apologize for the confusion and lack of clarity. We couldn’t agree more that no solid conclusion can be drawn with a very minimal number of controls. However, the small number of healthy subjects was included for the purpose of quality control for the NMR testing in order to establish a reference range of quantified lipoprotein subpopulations in the study cohort of T2DM patients. Unlike the very well-established conventional laboratory lipid panel reference ranges, not everyone is familiar with reference range for NMR-derived lipoprotein species. Levels of lipoproteins (size and concentration) are quite different among published studies, thus we’ve tested the 13 plasma samples of healthy control (random unidentified subjects) in the same condition to establish a more precise reference range, so we can appreciate potential changes of lipoprotein species in the T2DM group.

We have now deleted Figures 2-4 and ignored the statistical comparison between the reference group (healthy controls) and T2DM group. However, we have provided data of NMR-derived lipoprotein subpopulations of T2DM patients along with our internal reference values range (New Table 2).

We have also clarified the study design in the methods section. Our main focus was to assess the impact of diabetes on lipoproteins changes and to determine potential associations between plasma lipoprotein subpopulations and the emerging markers of insulin resistance (LPIR score) and inflammation (GlycA) among T2DM patients.

For results table 1 is not informative if data for control group is not represented. 

Thank you for highlighting this important point. As we mentioned above the small control group was randomly selected from unidentified healthy donors (no medical information was obtained) for the purpose of establishing the reference range for NMR-derived lipoprotein subpopulations.

Similarly for table 4 the comparison for sex differences is even more faulty because of the small sample size and lack of comparison. Stratified data in this case doesn't inform about the strength of association.

Thank you for this excellent suggestion. We agree that stratification of T2DM group in two groups based on the sex resulted in smaller subgroups that might not be informative enough about the strength of associations. We have now deleted Table 4 as suggested.

For discussion, authors are advised to elaborate more on the results of the study in comparison to other studies even if results are not in concordance. Almost all studies referenced agree with obtained results.

Thank you for raising this important point. We have now elaborated more on the results of our study taking in consideration as well relevant studies that are not in concordance. New references are added to the list highlighted. 

Reviewer 2 Report

The manuscript " Association of advanced lipoprotein subpopulation profiles with insulin resistance and inflammation in patients with type 2 diabetes mellitus" is aimed to o examine the effects of diabetes on plasma lipids including advanced lipoprotein subpopulations profiling and conventional lipid measurement in T2D patients. The authors concluded that monitoring insulin resistance-induced changes of lipoprotein subpopulations in T2D might help to identify novel biomarkers that could be useful for effective clinical intervention.

However, there are some issues to be corrected...

1.       Please, the expression is type 2 diabetes, not type -2 diabetes mellitus

2.       Please define aim of the study in the Abstract section

3.       Pleas add study design in the Abstract section  

4.       Please add information about the hospital (level of care)

Author Response

Responses to Reviewer’s #2 Comments

We thank the reviewer for his/her careful review and very useful comments/suggestions.

The manuscript "Association of advanced lipoprotein subpopulation profiles with insulin resistance and inflammation in patients with type 2 diabetes mellitus" is aimed to o examine the effects of diabetes on plasma lipids including advanced lipoprotein subpopulations profiling and conventional lipid measurement in T2D patients. The authors concluded that monitoring insulin resistance-induced changes of lipoprotein subpopulations in T2D might help to identify novel biomarkers that could be useful for effective clinical intervention.

However, there are some issues to be corrected...

  1. Please, the expression is type 2 diabetes, not type -2 diabetes mellitus

We have corrected this mistake.

  1. Please define the aim of the study in the Abstract section

Our main was to determine potential associations between plasma lipoprotein subpopulations and the emerging markers of insulin resistance (LPIR score) and inflammation (GlycA) among T2DM patients.

This is mentioned in the abstract and in the last paragraph of the introduction.

  1. Please add study design in the Abstract section  

We have included information about study design in the abstract.

  1. Please add information about the hospital (level of care)

Ministry of National Guard Health Affairs (MNGHA) is an internationally respected healthcare organization government-funded multispecialty accountable health system spread all over Saudi Arabia. It integrates clinical care, training, academics with research and cutting-edge technology for providing outstanding patient care. The MNGHA has the most developed health information system infrastructure globally, and it is the first organization to have reached the HIMSS INFRaM-6 maturity level.

Information about MNGHA hospital level of care is now included in the manuscript.

Reviewer 3 Report

The manuscript presents some limitations.

A limited literature review where the authors ignore part of previous publications, is a relevant problem. Parts of the discussions related to the results cannot be evaluated as the results are not clearly described. The discussion is poorly written and do not present results of other authors. How can you explain your results? Moreover, limitations of the study should be added. What is clinical significance of the study?

Please, grammatical errors must be revised.

Author Response

Responses to Reviewer’s #3 Comments

We thank the reviewer for his/her careful review and useful comments/suggestions.

The manuscript presents some limitations. A limited literature review where the authors ignore part of previous publications, is a relevant problem. Parts of the discussions related to the results cannot be evaluated as the results are not clearly described. The discussion is poorly written and do not present results of other authors. How can you explain your results?

We apologize for the lack of clarity in the discussion. We have now elaborated more by comparing and contrasting our results with published work by other investigators. Due to the type of manuscript (article), unlike review paper where we have the freedom to cite as many papers as we wish, we cannot cite all published papers on the topic. However, we have included new relevant references (list of citations increased by ~ 28%) and we hope no major manuscript is missing now. We would appreciate it if you could provide us with any specific relevant citations that might be missing, so we can include them. The new added citations are highlighted at the end of manuscript (references section).

Moreover, limitations of the study should be added. What is clinical significance of the study?

 A paragraph of study limitations was already included in the manuscript (just before conclusions section).

Clinical significance of the study: These findings may eventually help target intervention to individuals with high-risk for T2DM. The use of current conventional lipids panel might not fully offer an effective therapeutic for diabetic patients. Thus, identifying specific lipoprotein subpopulations that predict better insulin resistance and associated diabetic complications at early stages may help for implementation of effective preventive diagnosis and better patient management.

Please, grammatical errors must be revised.

As suggested, we have gone through the revised article and corrected minor typing mistakes and grammatical errors. Thank you!

Round 2

Reviewer 1 Report

Concerns had been addressed

Reviewer 3 Report

No futher comments